# Vardenafil Oral Dispersible Films (ODFs) with Advanced Dissolution, Palatability, and Bioavailability

**DOI:** 10.3390/pharmaceutics14030517

**Published:** 2022-02-26

**Authors:** Heba A. Abou-Taleb, Wesam W. Mustafa, Tarek Saad Makram, Lamiaa N. Abdelaty, Hesham Salem, Hamdy Abdelkader

**Affiliations:** 1Department of Pharmaceutics and Industrial Pharmacy, Faculty of Pharmacy, Merit University (MUE), Sohag 82755, Egypt; heba.ahmed@merit.edu.eg; 2Department of Chemical and Pharmaceutical Sciences, School of Life Sciences, Pharmacy and Chemistry, Kingston University London, Kingston upon Thames KT1 2EE, UK; 3Department of Pharmacy, Al-Mustafa University College, Baghdad 10064, Iraq; 4Department of Pharmaceutics and Industrial Pharmacy, Faculty of Pharmacy, October 6 University, Giza 12511, Egypt; tareksaadmakram@yahoo.com; 5Department of Clinical Pharmacy, Faculty of Pharmacy, October 6 University, Giza 12511, Egypt; lamiaa_nagy@yahoo.com; 6Department of Pharmaceutical Chemistry, Faculty of Pharmacy, Deraya University, New-Minia, Minia 61519, Egypt; hesham.salem@deraya.edu.eg; 7Department of Pharmaceutics, College of Pharmacy, King Khalid University, Abha 61441, Saudi Arabia; 8Department of Pharmaceutics, Faculty of Pharmacy, Minia University, Minia 61519, Egypt

**Keywords:** spontaneity, vardenafil, sorbitol, acesulfame, sucralose, oral dispersible films

## Abstract

Oral, quick response, and on demand, also known as a spontaneous oral treatment for erectile dysfunction, is highly needed by both patients and physicians. Vardenafil is selective (fewer side effects) and more effective in difficult-to-treat conditions than sildenafil. This study aims at fostering the dual objectives of using biomolecules such as artificial sweetening agents to solubilize and mask the bitterness of vardenafil loaded on biodegradable polymeric materials (PVA, MC, SA, and PVP K30) to fabricate oral, fast-dissolving films (vardenafil ODFs) in the mouth without the need for water to ingest the dosage form. Furthermore, coprecipitated-dispersed mixtures of vardenafil and three sweeteners (sorbitol, acesulfame K, and sucralose) were prepared and characterized using FTIR, DSC, and solubility studies. Moreover, eight different vardenafil ODFs were prepared using the solvent-casting method. Modified gustatory sensation test, in vitro disintegration, and release studies were performed. In addition, the optimized ODF (F8) was compared with the commercial film-coated tablets pharmacokinetically (relative bioavailability, onset, and duration of actions were estimated). The results indicated that the three sweetening agents had comparable solubilizing capacity. However, both sucralose- and acesulfame K-based ODFs have a more enhanced sweet and palatable taste than sorbitol-sweetened ODF. The SA- and PVP K30-based ODFs showed significantly faster disintegration times and release rates than MC. In conclusion, PVA has good film-forming properties, but a higher ratio of PVA adversely affected the disintegration and release characteristics. The % relative bioavailability for ODF was 126.5%, with a superior absorption rate constant (Ka) of 1.2-fold. The C_max_ and estimated T_max_ were compared to conventional film-coated tablets.

## 1. Introduction

Erectile dysfunction is among the most common health issues affecting men. The exact cause of erectile dysfunction is not well understood. However, many systemic issues and drug intake are associated with this male condition, such as hypertension, diabetes mellitus, consumption of beta-blockers, digoxin, anticancer, and antacid drugs (e.g., cimetidine) [1,2]. With the introduction of phosphodiesterase inhibitors (PDEI) in the late 1990s, the treatment of erectile dysfunction has been shifted dramatically to non-invasive oral therapy. Sildenafil (Viagra^®^ 100 mg and 50 mg film-coated tablets) was the first PDEI member launched into the market in 1998 [3]. A few years later, vardenafil was approved for the oral treatment of erectile dysfunction in 2003. PDEI oral therapy has substantially succeeded in treating erectile dysfunction, being non-invasive and convenient for the patients. However, delayed response, lack of spontaneity, and rapid sexual response have been reported with the film-coated tablets of vardenafil and sildenafil [4]. Both patients and clinicians demand fast and spontaneous responses to sexual intercourse [4]. Vardenafil is 5–10 times more potent and selective as a PDE inhibitor than sildenafil.

Furthermore, vardenafil has been reported to respond better in difficult-to-treat groups such as erectile dysfunction in diabetic patients [5]. More interestingly, phosphodiesterase 6 (photo-transduction in the eye) can be inhibited by large doses of sildenafil but less with vardenafil. Further, vardenafil is a more selective inhibitor of phosphodiesterase (PDE) 5 (the enzyme form presents in the corpus cavernosum) than other PDEs, especially PDE6 [6]. Inhibition of the PDE6 enzyme is involved in causing ocular side effects such as visual disturbance, raising intraocular pressure, and ischemic optic neuropathy [7]. However, these benefits can be offset by poor water solubility and variability in drug response and oral bioavailability with meals. The onset of action reported from vardenafil-coated tablets ranges from 30 to 60 min. This time can be doubled after ingestion of the tablet with heavy meals [8]. Delayed gastric emptying with meals and pH-dependent solubility of vardenafil are likely to be the main causes affecting clinical outcomes of vardenafil conventional tablets [9].

Oral dispersible films (ODFs) can solve variability in drug responses of some drugs with limited oral bioavailability. This is because the drug is ingested in a soluble form without the need for the tablets or capsules to disintegrate and dissolve drug particles in the gastrointestinal fluid. The ODFs are non-conventional solid dosage forms that are composed of polymeric films or strips that dissolve or disintegrate in a short period of time (a few minutes) in the oral cavity and mix with saliva with the convenience of not being chewed or requiring fluid to swallow a bulky tablet or capsule [10,11]. In addition, the ODFs can show advantages over a very close dosage form called oral disintegrating tablets (ODT). The ODFs are less likely to cause choking and shed fragments of undissolved particulate matter, as what occasionally happens with ODT.

Better efficacy, safety, and more reliable pharmacokinetics bypassing first-pass metabolism have also been reported with the ODFs [12,13]. Fast-dissolving, polymeric films of etoricoxib–cyclodextrin complexes were prepared with potential commercial viability [11]. The ODFs can be prepared by large-scale and straightforward techniques, such as the solvent-casting method, solid dispersion extrusion, and hot-melt extrusion [10]. Sweetening agents are essential excipients in formulating ODF to enhance palatability and taste, as well as speed up disintegration and dissolution of the film in the oral cavity and improve the mechanical properties of the ODFs [10,12]. Natural and artificial sweetening agents are water-soluble carriers and have diverse chemical structures. Our aim is to prepare solid complexes of the hydrophobic drug vardenafil and three selected sweetening agents (sorbitol, acesulfame potassium, and sucralose) with high sweetening power (their chemical structures are shown in Figure 1) and investigate their solubilizing and dissolution-enhancing capacities. Vardenafil–sweetening agent complexes will be loaded onto polymeric films to produce a fast onset of action for vardenafil. The effects of different sweetening agents on three different hydrophilic polymers (methylcellulose, sodium alginate, and polyvinyl pyrrolidone), commonly used to prepare polymeric films and other pharmaceuticals having good film-forming as well as disintegration properties, will be investigated. The effect of the film-forming polymer polyvinyl alcohol in different ratios on the disintegration time will also be studied.

## 2. Materials and Methods

Vardenafil HCl trihydrate (Batch number UT2210202) was given by El-Obour Modern Pharmaceutical Industries Co., Kaliobeya, Egypt. Sodium alginate (low viscosity grade) and glycerol were supplied by Loba Chemie Ltd., Mumbai, India. Polyvinyl pyrrolidone K30 and methylcellulose were purchased from Sigma Aldrich (St. Louis, MO, USA). Polyvinyl alcohol (molecular weight: 14,000) was purchased from BDH Chemicals Ltd., Poole, England. Sorbitol was supplied by Modernist Pantry, Eliot, ME, USA; acesulfame potassium was obtained from Sanofi Ingredients, Shanghai, China; sucralose was supplied by Asin Co., Mirkow, Poland.

### 2.1. Preparation of Physical and Solid Dispersion of Vardenafil: Sweetening Agents

Equal weights (150 mg each) of vardenafil and the three sweetening agents (sorbitol, sucralose, and acesulfame potassium) were weighed and mixed in a porcelain dish with a spatula for 3 min. The produced physical mixtures were stored in glass tubes until further use. To prepare the solid dispersed mixtures, vardenafil was dissolved in methanol (20 mL), and the sweetening agent was dissolved in distilled water (5 mL). The two solutions were mixed in a porcelain dish, stirred magnetically, and heated at 60 °C for complete evaporation. The solid mixtures were pulverized, passed through a sieve of pore size 100–125 µm, and stored in glass tubes until further use.

### 2.2. Solubility Studies

The solubility of vardenafil from a pure drug powder, physical mixtures, and solid dispersions in distilled water was studied in a shaking water bath (Jeio Tech, Daejeon, Korea) adjusted at 100 strokes per minute at 25 ± 1 °C for 48 h. Amounts of vardenafil equal or equivalent to 20 mg were added to 10 mL of distilled water. Samples (3 mL each) were filtered and determined for vardenafil by a UV-visible spectrophotometer (Shimadzu 16001, Kyoto, Japan) by measuring the absorbance at 250 nm.

### 2.3. FTIR and DSC Studies of the Prepared Solid Mixtures

Fourier-transform infrared (FT-IR)-generated spectra were obtained using the FTIR spectrophotometer (Shimadzu IR-345, Kyoto, Japan). Compressed potassium bromide discs composed of vardenafil, sorbitol, acesulfame potassium, and sucralose and their physical and dispersed mixtures were recorded in the range of 400 to 4000 cm^−1^.

Accurate weights (3–5 mg) of vardenafil, sorbitol, acesulfame potassium, and sucralose were placed in aluminum pans. The pan temperature was increased gradually from 30 to 400 °C at a rate of 10 °C/min using a differential scanning calorimeter (DSC) from Mettler Toledo Star System, Zurich, Switzerland.

### 2.4. Preparation of Oral Dispersible Film (ODF) of Vardenafil Using the Solvent Casting Method

The solvent castingt method was utilized to prepare the vardenafil ODFs. Different blends of polyvinyl alcohol (PVP) solution (5% *w/v*), methylcellulose (MC) solution (2% *w/v*), sodium alginate (SA) solution (5% *w/v*), and polyvinylpyrrolidone K 30 (PVP) solution (2% *w/v*) were investigated, as shown in Table 1. Glycerin (plasticizer) was added at a concentration of 33% *w/w* of the dry weight. Dispersed mixtures of vardenafil and the sweetening agents (equal weight) were dissolved in the final polymer solution to make a concentration of 10% *w/w* of the dry weight of the total polymers and plasticizer used. According to the composition mentioned in Table 1, different film formulations were mixed in a small beaker (50 mL) and poured into a petri dish (9 cm in diameter). The Petri dishes were allowed to dry in an ambient and well-ventilated area. The casted film was cut out using a cork borer (1.5 cm in diameter).

### 2.5. Characterization of Vardenafil ODFs

#### 2.5.1. Thickness, Weight, Drug Content, and Surface pH

For each formulation, six different vardenafil ODFs were selected, and individual thickness and weight were measured using a digital micrometer and an analytical balance (Mettler Toledo, Zurich, Switzerland). The mean value was estimated and presented as the mean ± standard deviation (SD). Six vardenafil ODFs were individually dissolved in 10 mL of hydroalcoholic solution (50%), filtered and adequately diluted with distilled water, and measured spectrophotometrically, as described in Section 2.2.

The surface pH of the prepared vardenafil ODFs was measured, as reported elsewhere [14]. A volume of 0.5 mL was dropped over the vardenafil ODFs and allowed to wet and hydrate the ODFs. The pH was recorded using pH indicator strips (VWR International Ltd., Poole, UK).

#### 2.5.2. Folding Endurance

Four vardenafil ODFs were folded repeatedly at the center until the film broke. The number of times that the ODFs were folded until rupture was the value of folding endurance. The means of six measurements of six different films were used (±SD).

#### 2.5.3. In Vitro Disintegration Time

Six vardenafil ODFs were placed individually in 6 glass tubes of the basket assembly immersed in a thermo-stated water bath containing 1 L of phosphate buffer (pH 7) at 37 ± 1 °C using Erweka ZT 121, GmbH, Langen, Germany. The basket assembly was raised and lowered at a frequency of 30 cycles per min. The time required to disintegrate or break up was taken as the disintegration time.

#### 2.5.4. Modified Gustatory Sensation Test

The palatability of the prepared vardenafil ODFs was assessed using the human gustatory test [15]. Six adult volunteers (aged 25–40 years old) were recruited for this study. This study was approved by the Ethics Approval Committee (Code Number: HV04/2021) and approved by the Faculty of Pharmacy, Minia University. For bitterness assessment, quinine HCl solutions of increasing concentrations of 0.001, 0.01, 0.03, 0.1, 0.3, and 1.0 mM were used as standard solutions with bitterness scores of 0 (no bitter/tasteless), 1, 2, 3, 4, and 5 (extremely bitter), respectively [15]. For the degree of sweetness, syrups (68% sugar solution) and three other dilutions—1:1, 1:2, and 1:3—were given sweetness scores of −4, −3, −2, and −1, respectively. The volunteers were instructed to place a sample volume (1 mL) of the standard quinine solutions or syrup solutions onto the middle of the tongue and allowed them to stay in the mouth for ten seconds, after which they were asked to write down their bitterness and sweetness scores. After 1 h, every volunteer was allowed to place one vardenafil ODF on the tongue and keep it until disintegration (up to 10 min), and the volunteers were asked to tell the ODFs scores. The volunteers were asked to provide a bitterness intensity score for the samples. A washout period of 20 min was allowed between each ODF, where the volunteer was asked to gargle with plenty of water.

#### 2.5.5. In Vitro Release Study

In vitro vardenafil release from the prepared ODFs was studied using USP dissolution apparatus 2. The release medium was 500 mL of phosphate buffer (pH 7) containing 0.1% tween-80. The temperature was set at 37 ± 1 °C and stirred with the paddle at 50 rpm. A sample (5 mL) was withdrawn at specified intervals of 5, 10, 15, 30, and 60 min. The drug content was determined spectrophotometrically, as mentioned in Section 2.2.

### 2.6. Pharmacokinetics and Blood Pressure Monitoring Studies

Six male volunteers (aged 26–45 years old) were recruited for this study based on the Ethics Approval (Code Number: HV07/2020) approved by the Faculty of Pharmacy, Minia University. The six volunteers were fasted (to eliminate inter-subject variability due to food) and divided into two groups (3 volunteers each). Group 1 received a single dose (10 mg) of film-coated tablets of vardenafil (Vardapex^®^ film-coated tablet). Group 2 received a single dose (10 mg) of optimized vardenafil ODF (F8). Blood pressure for all volunteers was monitored during the whole study. Blood samples (2 mL each) were collected at time intervals of 0.25, 0.5, 1, 4, 8, and 24 h. Plasma was separated by centrifugation of the blood samples at 1000 g for 10 min. The samples were stored at −20 °C till the HPLC assay.

#### HPLC Analysis of Vardenafil in Plasma and Pharmacokinetics Data Processing

A total of 1 mL of human plasma was added to 2.5 mL of the standard drug; the spiked plasma was blended with 6 mL of acetonitrile for deproteinization. The blend was centrifuged at 3000 rpm for 10 min. The supernatant (plasma) was poured into a 50 mL volumetric flask, and the volume was brought to the mark with the mobile phase. The HPLC system was a Dionex UltiMate 3000 RS system (Thermo Scientific, Dionex Sunnyvale, CA, USA) and an Inertsil^®^ ODS-3 (25 cm × 0.46 cm, 5μm) column kept at 25 °C. The mobile phase consisted of water with pH 2.6 (pH adjusted using phosphoric acid) and methanol at a 40:60 *v*/*v* ratio. The flow was isocratic at a rate of 1 mL/min. The detection was performed at 254 nm, and an injection volume was set at 20 μL.

The plasma concentrations versus time curves for Vardapex^®^ film-coated tablets and vardenafil ODFs were constructed, and the data were processed using PK Solver software.

### 2.7. Statistical Analysis

The unpaired *t*-test and one-way ANOVA followed by Dunnett’s multiple comparisons test were performed using GraphPad Prism version 8.4.3 software, San Diego, CA, USA.

## 3. Results and Discussion

Three different water-soluble and non-cariogenic (they do not cause tooth decay) sweetening agents were studied. Acesulfame potassium (acesulfame K) and sucralose are 200 and 600 times sweeter than sucrose (the table sugar), respectively. Sorbitol’s sweetening power is much less compared to sucralose and acesulfame K, approximately 60% of sucrose [16]. In this study, the solubilizing and dissolution-enhancing effects of these three sweeteners were investigated with the poorly soluble drug vardenafil in addition to their primary use to enhance the organoleptic properties of vardenafil ODFs.

### 3.1. Solubility Studies

Solubility of vardenafil, vardenafil:sorbitol, vardenafil:acesulfame K, and vardenafil:sucralose PM and Coppt are summarized in Figure 2. The solubility of vardenafil in water was found to be 0.4 mg/mL. Enhancement of vardenafil solubility was recorded for vardenafil:sorbitol, vardenafil:acesulfame K, and vardenafil:sucralose PM and Coppt. The solubility of vardenafil increased up to 0.7 mg/mL by 1.75-fold. Superior enhancement was recorded for sucralose. The ranking of solubilizing capacity was in the following order: sucralose > acesulfame K > sorbitol. It is worth mentioning that the preparation technique markedly affected the solubilizing capacity; the coprecipitated mixtures showed better solubility compared to its physical mixture counterparts. Coprecipitation and solid dispersion could bring more intimacy and physical interactions between vardenafil and the water-soluble carriers compared to physical mixing of the powders without prior solubility in a common solvent.

Similar results were reported elsewhere with different excipients. Enhancement of vardenafil solubility was also reported with other hydrophilic carriers such as hydroxypropyl methyl cellulose (HPMC) and β-cyclodextrin, Complete amorphization of vardenafil was obtained at 1:1 and 1:5 weight ratios using the freeze-drying technique, improving solubility by 1.44-fold [17].

### 3.2. FTIR and DSC Studies

Figure 3, Figure 4 and Figure 5 show the FTIR spectra with embedded chemical structures of vardenafil, sorbitol, vardenafil:sorbitol PM and Coppt; vardenafil, acesulfame K, vardenafil: acesulfame K PM and Coppt; and vardenafil, sucralose, vardenafil:sucralose PM and Coppt; respectively. The main characteristic FTIR peaks of vardenafil at 3410, 2900, 1720, 1650, and 1600 cm^−^^1^ were assigned for -NH (stretching), -CH_2_**/**-CH_3_ (stretching), C=O (stretching), C-H (bending), and C=C (aromatic stretching), respectively [9]. Sorbitol characteristic FTIR peaks included a broad stretching absorption peak from 3200–3500 cm^−^^1^ for O-H (alcohol), and alkyl (C-H, CH_2_) peaks appeared at 2850 cm^−^^1^ (Figure 3). Acesulfame K characteristic FTIR peaks included weak broad stretching absorption peaks for C-H, CH_3_, C=O, and C=C at 3250–3500 cm^−^^1^, 1680, and 1620 cm^−^^1^, respectively (Figure 4). Sucralose characteristic FTIR peaks included C-H and OH (alcohol), which appeared at 3250–3500 cm^−^^1^ (Figure 5). Physical mixtures of vardenafil with the three sweetening agents (sorbitol, acesulfame K, and sucralose) were simple superimposition of the peaks assigned for the vardenafil and corresponding sweetening agent. For vardenafil:sorbitol Coppt, broadening of the single primary amine peak (N-H) and integration into the broad peak from 3000–3500 cm^−^^1^ could be ascribed to the formation of H-bonding between the hydroxyl groups (OH) of sorbitol and -NH of vardenafil as a result of the preparation by the coprecipitation technique. Similarly, vardenafil:acesulfame K and vardenafil:sucralose showed similar behavior (Figure 4 and Figure 5). These weak electrostatic interactions (H-bonding) could explain the recorded enhancement of vardenafil solubility from the prepared coprecipitated dispersion with three sweetening agents.

DSC thermograms of vardenafil, sorbitol, vardenafil:sorbitol PM and Coppt; vardenafil, acesulfame K, vardenafil:acesulfame K PM and Coppt; and vardenafil, sucralose, vardenafil:sucralose PM and Coppt, are recorded in Figure 6, Figure 7 and Figure 8, respectively.

The thermogram of vardenafil showed a broad endothermic peak from 50–100 °C, and another sharp endothermic peak appeared at 238 °C, corresponding to water loss of hydration and vardenafil melting, respectively. Sorbitol showed a sharp endothermic peak at 98 °C; this is attributed to sorbitol melting. Another broad peak from 280–320 °C could be ascribed to the charring and decomposition of sorbitol. Complete disappearance of the vardenafil melting peak with vardenafil:sorbitol PM and Coppt was observed (Figure 6). Acesulfame K showed an exothermic peak at 230 °C, indicating melting with decomposition. Both vardenafil:acesulfame K PM and Coppt showed broad peaks at 50 to 100 °C, indicating loss of water of hydration (free moisture) and complete disappearance of the vardenafil melting peak (Figure 7). Sucralose showed a sharp endothermic peak at 120 °C, indicating sucralose melting. The vardenafil:sucralose PM showed a marked shift of the vardenafil endothermic peak to a lower melting peak at 140 °C. This indicates physicochemical interactions and reduced crystallinity of vardenafil. Complete disappearance of the drug peak, indicating complete amorphization of vardenafil with sucralose, could be correlated well with the superior solubility of vardenafil compared to the other sweetening agents used.

### 3.3. Preparation of Vardenafil ODFs

Eight different vardenafil ODFs formulations (F1–F8) were prepared using the solvent-casting method. All prepared ODFs formulations contained 10% *w/w* of vardenafil; F1 represented a control formulation without sweetening agent, and F2 sorbitol, F3 acesulfame K, and F4 sucralose (Table 1). PVA, a film-forming agent, and three different polymers, MC, SA, and PVP K 30, were investigated at different ratios of PVA:MC:SA:PVP K30 as follows: 1:1:1:0 for F1 to F4; 0.5:1:1.5:0 for F5 (acesulfame K) and F6 (sucralose) and 0.5:0:1.5:1 for F7 (acesulfame K) and F 8 (sucralose).

### 3.4. Characterization of the Prepared Vardenafil ODFs

Various parameters were measured to evaluate and characterize the prepared vardenafil ODT, such as thickness, weight, drug content, surface pH, folding endurance, and in vitro disintegration (Table 2). The thickness was less than 1 mm and ranged from 0.34 to 0.40 mm. The weight of the prepared vardenafil ODFs ranged from 31.5 to 38.5 mg. The average vardenafil content in the prepared ODFs formulation ranged from 9.0 ± 0.2 to 10 ± 0.3 mg with the percentage (%) drug content ranging from 90 to 100%, indicating acceptable and uniform drug content in the prepared vardenafil ODFs. The measured surface pH was in the range of 7.0 to 7.5. The film’s mechanical properties recorded folding endurance 10 ± 2.5 to 20 ± 3, indicating good elasticity and tensile strength that the prepared vardenafil ODFs formulation could withstand mechanical stress due to manufacturing and patients’ handling. Glycerin is an efficient plasticizer for flexible and malleable film formation. The lowest folding endurance value (10 ± 2.5) was recorded for F1 (no sweetening agent). These results indicate that the used sweetening agents have an additional plasticizing capacity, and they could contribute to producing additional plasticizing properties. Similar results were reported elsewhere for the plasticizing characteristics of sorbitol. The hydroxyl (-OH) groups of sorbitol can form H-bonding with polyvinyl alcohol and help retain water molecules [18]. Similarly, the -OH groups of the sucralose molecules could do so. This could explain the superior mechanical properties of F2 (sorbitol) and F4 (sucralose), compared with F1(no sweetening agent) and F3 (acesulfame K).

In vitro disintegration was performed in phosphate buffer (pH 7) at 37 °C, simulating saliva [19]. The recorded in vitro disintegration time was 2.0 ± 1.0 to 16 ± 2.5 min. Both the composition of the film and the type of sweetening agent seem to be significant contributing factors. F1 showed the longest disintegration time compared with other ODFs. F1 did not contain a sweetening agent and contained the highest concentration of polyvinyl alcohol. Addition of sweetening agents in F2–F4 significantly (*p* < 0.05) shortened the disintegration time by up to half (8 min) with F4. Sweetening agents are highly water-soluble and quickly dissolve, creating pores and channels that could facilitate the disintegration of the ODFs. The disintegration time significantly (*p* < 0.001) decreased by approximately 50% when the PVA decreased by the same percentage (50%) (Table 1). For example, F3 (11 min) and F5 (7 min) contained PVA:MC:SA at a ratio of 1:1:1 and 0.5:1:1.5, respectively. Further, replacement of MC in F6 with PVP K30 in F8 produced vardenafil ODFs formulations with a significantly shorter (*p* < 0.05) and more acceptable disintegration time (2 min).

Moreover, the viscosity of the polymer can inversely affect the disintegration time. PVP is a highly water-soluble polymer with a lower viscosity grade than MC. The lower the viscosity of the polymer, the shorter the disintegration time of ODFs could be [19]. These results could indicate that the presence of SA and PVP could have better disintegration properties than MC and PVA. Sucralose seems to be superior among the three used sweetening agents.

### 3.5. Modified Gustatory Sensation Test

Bitterness scores and sweetness scores were recorded for the prepared eight vardenafil ODFs formulations, as shown in Figure 9. F1 recorded the highest bitterness score comparable to the extremely bitter reference standard concentration (approximately 4). This can be attributed to the fact that vardenafil is a bitter-tasting drug. F2 (sorbitol) showed a moderate bitterness score of 2.75. This indicates that sorbitol in the 1:1 *w/w* ratio was not sufficient to mask the unpalatable taste of vardenafil due to the lowest sweetening power. On the contrary, both acesulfame K- and sucralose-sweetened vardenafil ODFs (F3–F8) recorded a sweet taste, as shown in Figure 9. Sucralose-sweetened ODF formulations (F4, F6, and F8) recorded higher sweetness scores than the acesulfame K-sweetened ODFs (F3, F5, and F7). This correlated well with the sweetening capacity of the sweetening agents used, namely, sucralose > acesulfame K >> sorbitol. Therefore, vardenafil ODFs from F3 to F8 were selected for further dissolution studies.

### 3.6. In Vitro Drug Release

In vitro vardenafil release from the prepared eight ODFs formulations is shown in Figure 10. The release medium simulates saliva fluid (pH 7 at a temperature of 37 °C). F1 recorded the slowest release rate compared to all other ODFs formulations. For example, complete vardenafil release (100%) from F1 was attained at 60 min; in contrast, the fastest vardenafil release rates (100% drug release per 10 min) were recorded for both F7 and F8. While F2–F6 recorded relatively moderate drug release rates, complete drug release rate was attained at 30 min for F2 and 15 min for F3–F5. The release rates were correlated well with the in vitro disintegration times recorded for the prepared vardenafil ODF formulations and the viscosity grades of the polymers used to fabricate the oral films. Lower viscosity-grade polymers, the SA- and PVP K30-based ODFs, demonstrated faster disintegration times and release rates.

Further, the presence of sweetening agents contributed significantly (*p* < 0.05) to increasing release rates. For example, F1 (no sugar) showed doubled the time (60 min) required for complete drug release compared to that (30 min) of F2 (sorbitol). This can be ascribed to better solubilization and enhanced dissolution from the vardenafil-sweetening agent dispersed mixture. Sorbitol and the other sweetening agents used have a high water solubility that can dissolve faster and create channels and pores in the ODFs, allowing water molecules to penetrate and dissolve the polymeric films faster.

### 3.7. Pharmacokinetics Study and Blood Pressure Monitoring

The plasma concentrations of the vardenafil–time curves of the Vardapex^®^ (10 mg) film-coated tablets and vardenafil (10 mg) ODF formulations are shown in Figure 11. From the plasma curves, the faster absorption (steep) phase and greater overall area under the curve (AUC) for the ODFs were markedly more recorded than for the film-coated tablets. This can be ascribed to the tested ODFs’ fast dissolution and absorption rate. Table 3 shows the calculated pharmacokinetic parameters for the Vardapex^®^ film-coated tablets and the optimized vardenafil ODF (F8), estimated from plasma concentration–time curves using the PK Solver software. The estimated parameters—Ka, K, C_max_, T_max_, MRT0-∞, T_1/2_, and AUC0-24, were the absorption rate constant, elimination rate constant, peak plasma concentration, time to reach the peak plasma concentration, mean residence time, plasma half-life, and area under the curve, respectively.

Both the absorption rate (Ka) constant and extent (as estimated from AUC) for vardenafil ODFs were significantly higher (*p* < 0.05) than those recorded for the film-coated tablet. The maximum plasma concentration (C_max_) recorded for vardenafil ODFs was 1.3-fold greater than that recorded for the commercial tablet. Similarly, the AUC for the tested ODFs recorded the same increment.

The improvement in vardenafil bioavailability recorded for the vardenafil ODFs could be ascribed to the solubilization of the drug by sucralose in the oral film and having a superior disintegration and dissolution rate in the buccal cavity. Therefore, drug molecules reach absorption sites in the gastrointestinal tract (GIT) ready for absorption, compared with the more time required for disintegration and dissolution of the film-coated tablet in the GIT fluid. On the other hand, other pharmacokinetic parameters, such as the elimination rate constant (K) and plasma half-life (T_1/2_), measured for the Vardapex^®^ film-coated tablets and F8 showed no statistically significant differences (*p* > 0.05). These two parameters are mainly dependent on drug distribution, metabolism, and excretion [20]. Whilst these latter bioprocesses were not altered markedly by presenting the same drug in either oral tablet or oral disintegrating films, this could explain the non-significant differences between the two pharmacokinetic parameters (K and T_1/2_) recorded.

It is worth mentioning that blood pressure measurements and pulse rates did not show statistical significance between the two groups (Vardapex^®^ film-coated tablets and vardenafil ODFs) in terms of lowering the blood pressure of the recruited volunteers. This can be attributed to the fact that vardenafil is a selective and potent PDE5 inhibitor and is less likely to cause cardiovascular events than other members such as sildenafil [7].

## 4. Conclusions

There are many reasons for selecting vardenafil, among other PDE inhibitors, to be loaded on orally dispersible films. Vardenafil is widely prescribed treatment for erectile dysfunction; it is a more potent and selective inhibitor of PDE5. Furthermore, vardenafil is more effective in difficult-to-treat cases with fewer visually disturbing unwanted side effects. The results showed that vardenafil had an extremely bitter taste and erratic absorption due to poor solubility. In addition, the delayed onset and erratic oral absorption from film-coated tablets due to poor solubility and food effects represent the main drawbacks of vardenafil penile therapy.

An efficient approach was investigated to generate drug-sweetening agent complexes that were more soluble than the drug alone. Vardenafil–sweetening agent complexes were successfully loaded onto polymeric films that can quickly dissolve in the mouth to produce rapid plasma peaking for a relatively extended period compared to that of film-coated tablets. Moreover, sucralose seems to be a superior solubilizing and sweetening agent than acesulfame K and sorbitol. Additionally, combinations of sodium alginate (SA) and polyvinyl pyrrolidone (PVP) K 30 exhibited a shorter disintegration time in vitro and faster dissolution rates than other vardenafil ODFs. In conclusion, personalized, more convenient, and more effective responses than conventional film-coated tablets of the phosphodiesterase inhibitor vardenafil can be achieved through ODF systems.

## Figures and Tables

**Figure 1 pharmaceutics-14-00517-f001:**
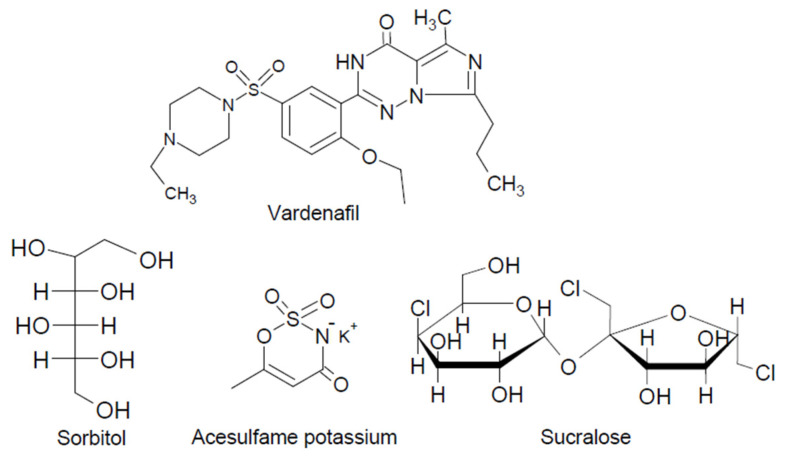
Chemical structures of the three sweetening agents and vardenafil.

**Figure 2 pharmaceutics-14-00517-f002:**
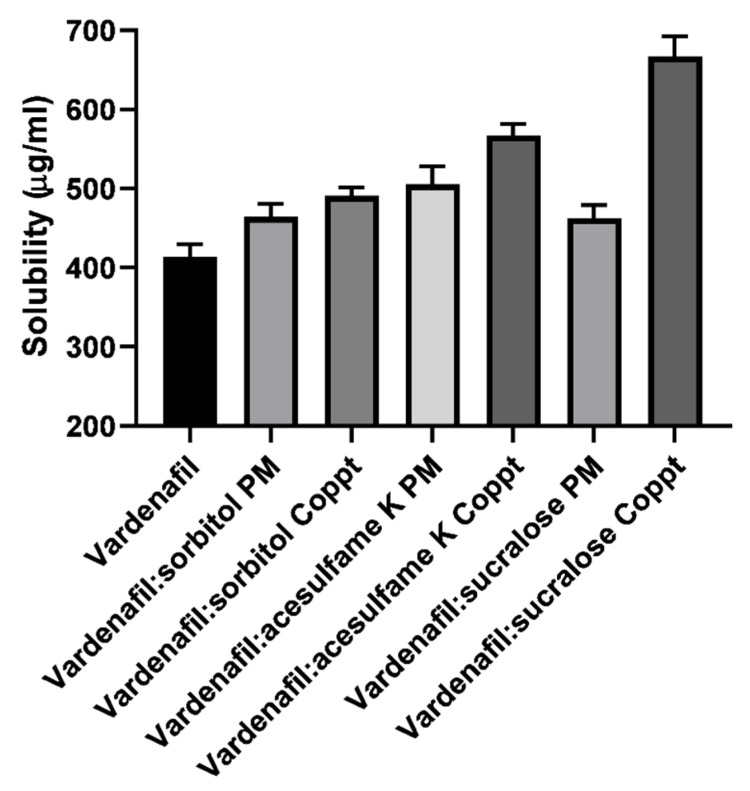
Solubility of vardenafil, vardenafil:sweetening agents physical (PM) and dispersed (Coppt) mixtures.

**Figure 3 pharmaceutics-14-00517-f003:**
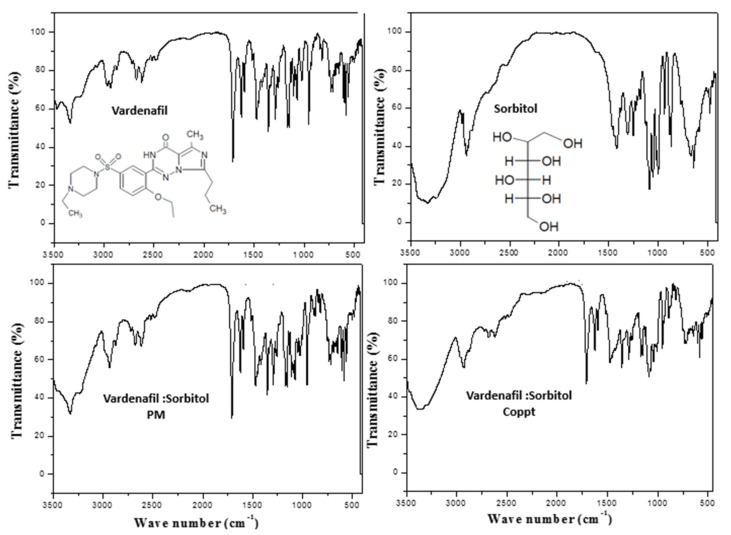
FTIR spectra of vardenafil, sorbitol, vardenafil:sorbitol PM, and vardenafil:sorbitol Coppt.

**Figure 4 pharmaceutics-14-00517-f004:**
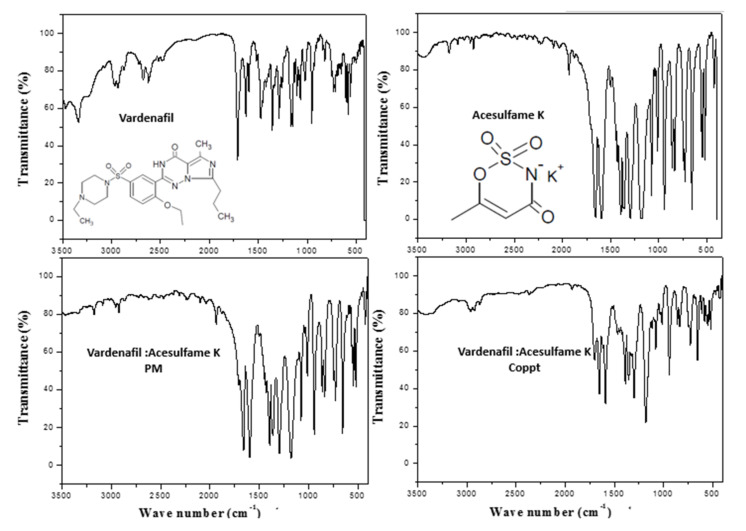
FTIR spectra of vardenafil, acesulfame K, vardenafil:acesulfame K PM, and vardenafil:acesulfame K Coppt.

**Figure 5 pharmaceutics-14-00517-f005:**
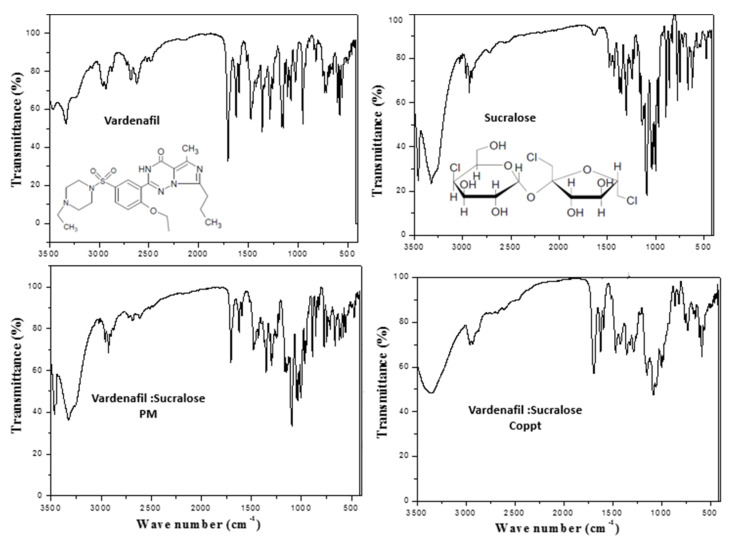
FTIR spectra of vardenafil, sucralose, vardenafil:sucralose PM, and vardenafil:sucralose Coppt.

**Figure 6 pharmaceutics-14-00517-f006:**
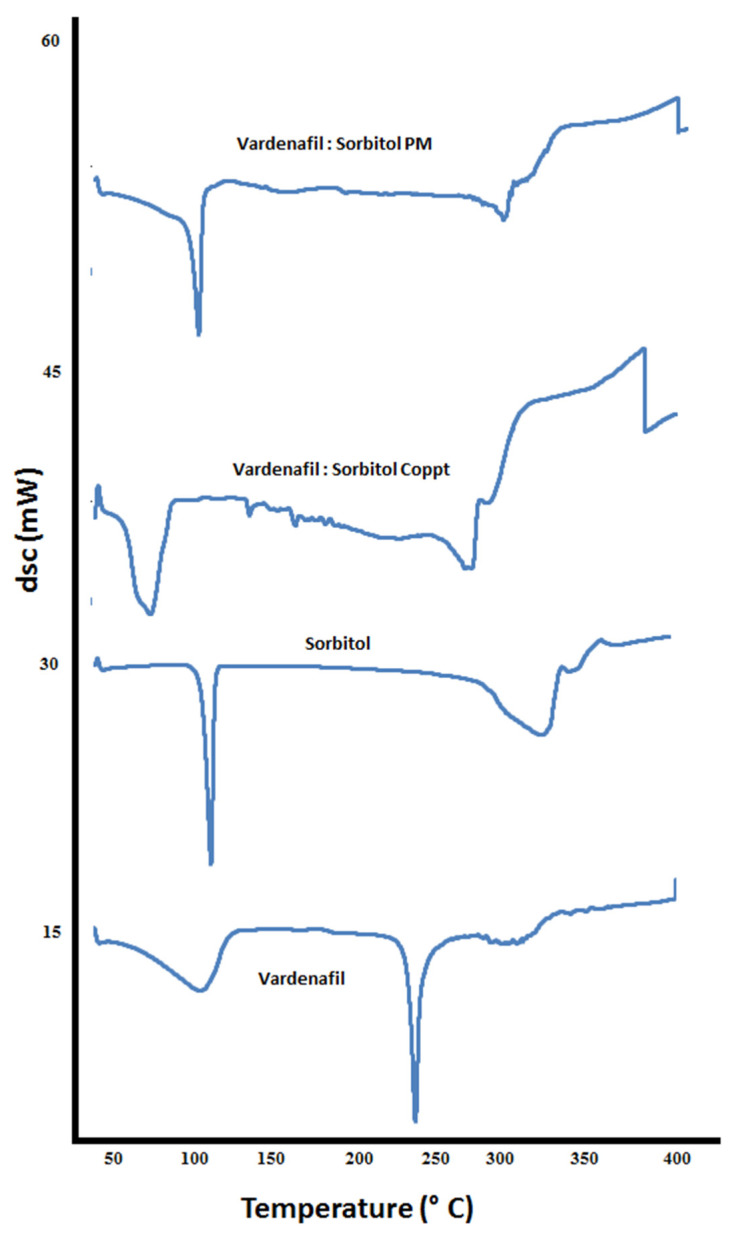
DSC thermograms of vardenafil, sorbitol, vardenafil:sorbitol PM, and vardenafil:sorbitol Coppt.

**Figure 7 pharmaceutics-14-00517-f007:**
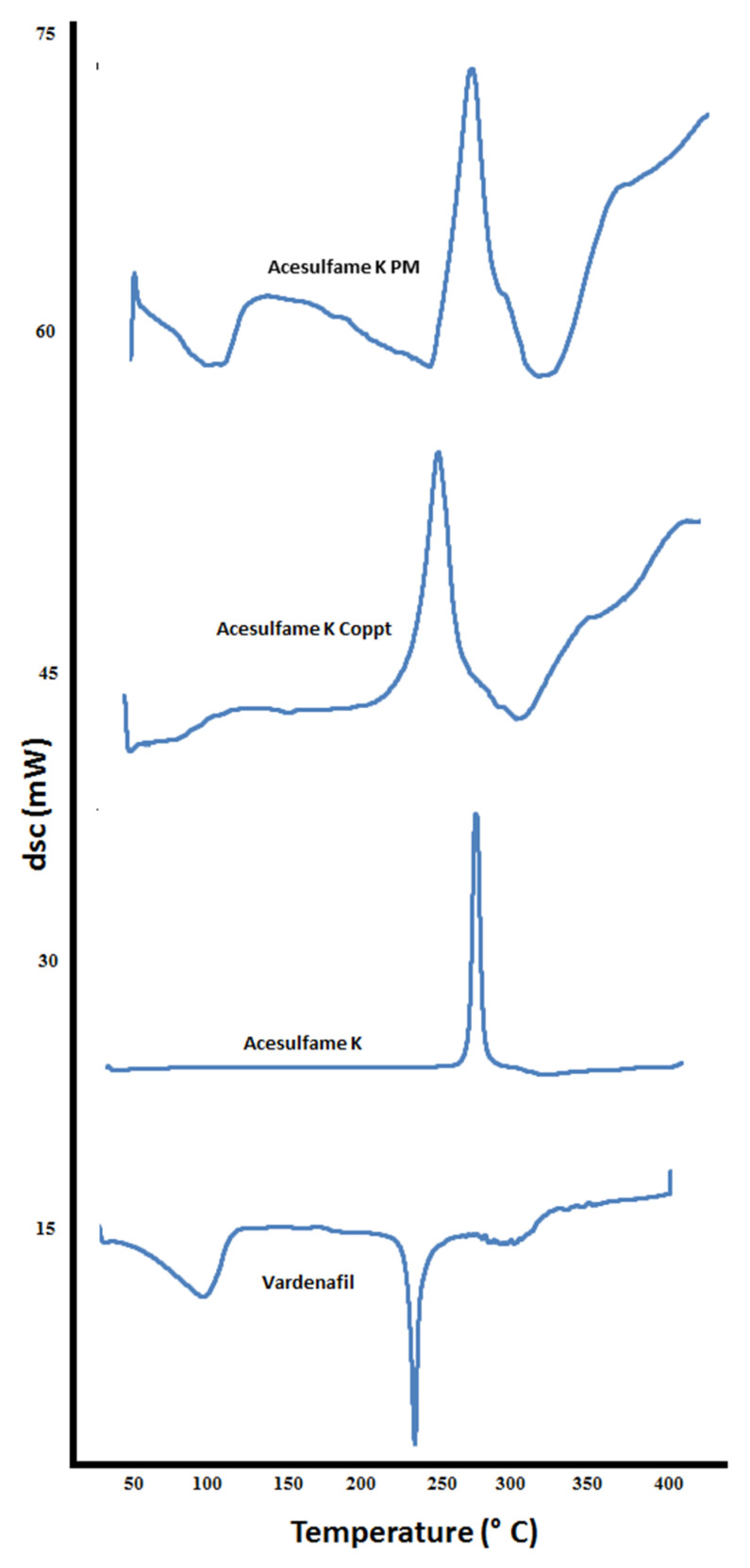
DSC thermograms of vardenafil, acesulfame K, vardenafil:acesulfame K PM, and vardenafil:acesulfame K Coppt.

**Figure 8 pharmaceutics-14-00517-f008:**
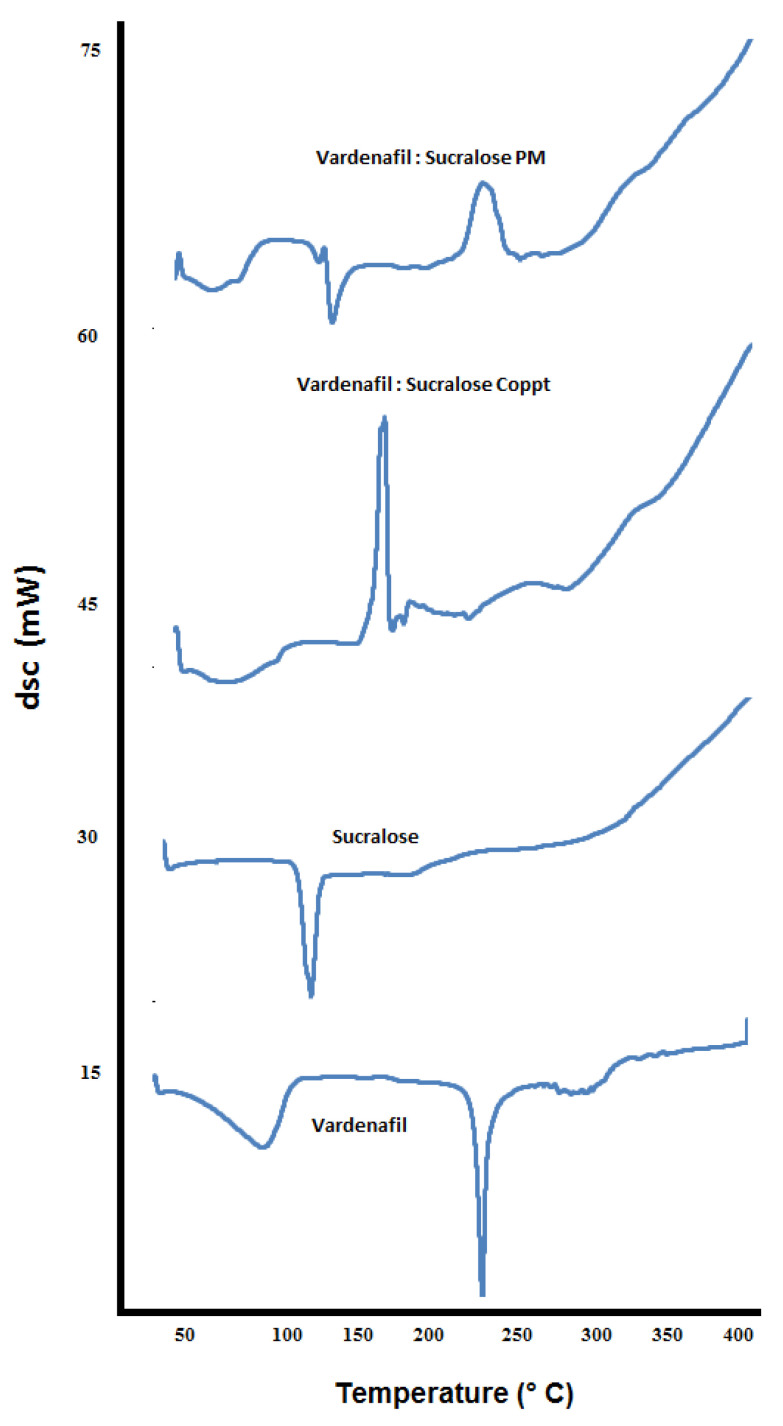
DSC thermograms of vardenafil, sucralose, vardenafil:sucralose PM, and vardenafil:sucralose Coppt.

**Figure 9 pharmaceutics-14-00517-f009:**
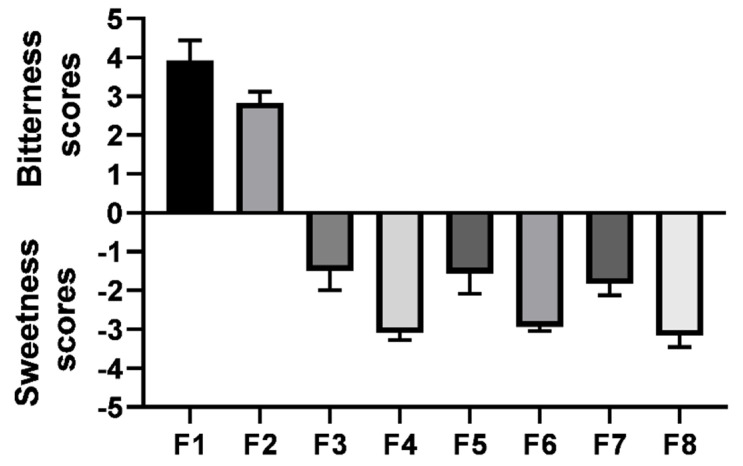
Bitterness (positive Y-scale) and sweetness scores (negative Y-scale) of the prepared vardenafil ODFs formulations (data represented as the mean ± SD).

**Figure 10 pharmaceutics-14-00517-f010:**
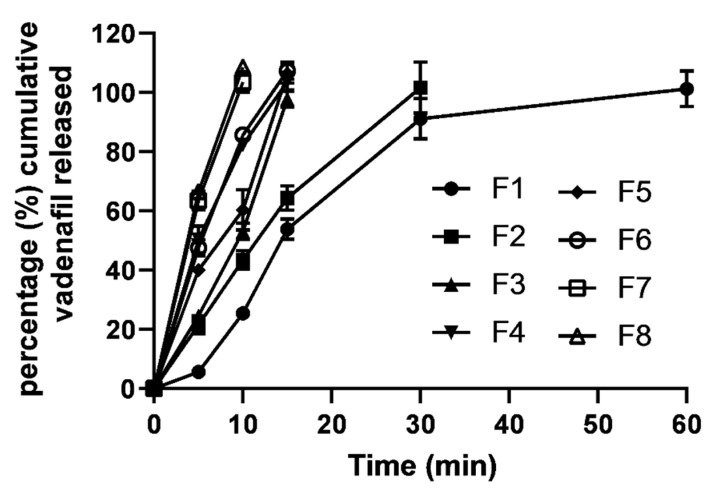
In vitro drug release profiles of vardenafil from the prepared ODF formulations.

**Figure 11 pharmaceutics-14-00517-f011:**
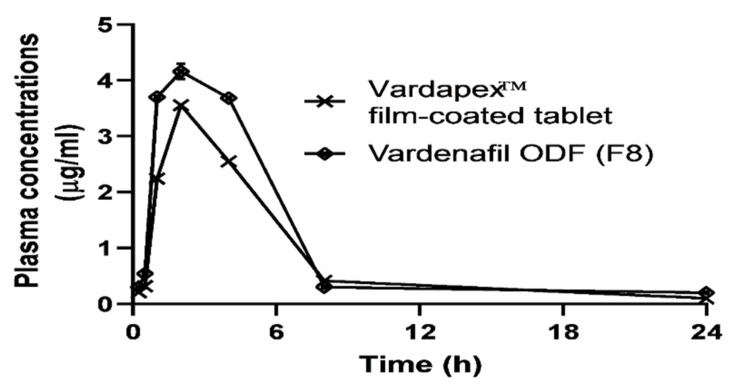
Plasma concentrations of the vardenafil–time (h) curves of commercially available film-coated tablets (Vardapex^®^ 10 mg) and the optimized vardenafil ODF (F8), from the recruited human volunteers. Data points represent the mean ± SD (*n* = 3).

**Table 1 pharmaceutics-14-00517-t001:** Compositions and formulation symbols of the vardenafil oral dispersible films (ODFs) formulations.

Formulation Symbol	F1	F2	F3	F4	F5	F6	F7	F8
Ingredients								
Vardenafil (mg)	120	120	120	120	120	120	120	120
Sorbitol (mg)	-	120	-	-	-	-	-	-
Acesulfame potassium (mg)	-	-	120	-	120	-	120	-
Sucralose (mg)	-	-	-	120	-	120	-	120
PVA 5% *w/w* (g)	10	10	10	10	5	5	5	5
MC 2% *w/w* (g)	10	10	10	10	10	10	-	-
SA 5% *w/w* (g)	10	10	10	10	15	15	15	15
PVP K30 2% *w/w* (g)	-	-	-	-	-	-	10	10
Glycerin (mg)	435.6	435.6	435.6	435.6	435.6	435.6	435.6	435.6

Note: Vardenafil concentration was calculated based on the percentage of the solid weight of the polymers and kept at a concentration of 10% *w/w*. PVA, MC, SA, and PVP K30 denote polyvinyl alcohol, methylcellulose, sodium alginate, and polyvinylpyrrolidone K30, respectively.

**Table 2 pharmaceutics-14-00517-t002:** Dimensions (thickness and weight), drug content, surface pH, folding endurance, and in vitro disintegration time for the vardenafil ODFs (data presented as the mean ± SD).

Formulation Symbol	F1	F2	F3	F4	F5	F6	F7	F8
Thickness (µm)	400 ± 5.0	380 ± 4.5	390 ± 3.5	385 ± 5.8	360 ± 8.5	365 ± 8.5	340 ± 3.0	350 ± 3.5
Weight (mg)	31.5 ± 0.3	38.5 ± 3.3	37.5 ± 2.3	38 ± 2.5	36 ± 3.4	37 ± 2.7	38 ± 2.8	38 ± 1.45
Drug content (mg)	9.5 ± 0.5	9.0 ± 0.2	9.8 ± 0.6	8.9 ± 0.5	9.0 ± 0.3	9.4 ± 0.4	10 ± 0.3	9.5 ± 0.2
Surface pH	7.5	7.0	7.0	7.5	7.5	7.0	7.0	7.0
Folding endurance	10 ± 2.4	15 ± 2.0	12 ± 2.5	20 ± 2.6	10 ± 2.1	16 ± 2.0	12 ± 2.5	14 ± 2.0
In vitro disintegration time (min)	16 ± 2.5	11.0 ± 2.1	11 ±1.5	8.0 ± 1.0	7.0 ± 1.0	5.0 ± 1.0	3.0 ± 1.5	2.0 ± 1.0

**Table 3 pharmaceutics-14-00517-t003:** Pharmacokinetics parameters estimated for vardenafil film-coated tablet and vardenafil ODF (F8) from human plasma. Data represent the mean ± SD (*n*= 3).

Pharmacokinetics Parameters	Vardapex^®^ (10 mg) Film-Coated Tablet	Vardenafil ODF (F8)
Ka (min^−1^) *	0.0073 ± 0.0001	0.0085 ± 0.00027
K (min^−1^) **	0.0065 ± 0.0003	0.0066 ± 0.00027
Observed T_max_ (min) **	120	120
Estimated T_max_ (min)	144 ± 1.0	130 ± 1.0
C_max_ (µg.ml^−1^) *	3.56 ± 0.08	4.61 ± 0.11
MRT_0-ꝏ_ (min)	340.9 ± 7.6	388.2 ± 6.8
T_1/2_ (min)	268.8 ± 6.9	317.4 ± 8.2
AUC_0-t_ (µg.ml^−1^. min) *	1188 ± 15.87	1499 ± 24.70
% Relative bioavailability ***	-	126.5 ± 3.04

* Denotes a statistically significant difference (*p* < 0.05). ** Observed T_max_ was obtained from the time to peak plasma concentration; estimated T_max_ was calculated from PK solver software after constructing the log plasma concentration versus time curve by adopting the residual and feathering technique. *** % Relative bioavailability was the total AUC0-24 for F8 divided by AUC0-24 for the conventional film-coated tablet and normalized to 100.

## Data Availability

Data available within the article.

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
