# Peer review of "Vardenafil Oral Dispersible Films (ODFs) with Advanced Dissolution, Palatability, and Bioavailability"

_pharmaceutics, 2022, doi:10.3390/pharmaceutics14030517_

Round 1

Reviewer 1 Report

The manuscript is well structured and organized. The solubility and in turn bioavailability of vardenafil was improved by making complexes of vardenafil with three sweetening agents. The lab scale findings could be used for future prospective to go in this direction. The results are systematically presented and well narrated. However, few formatting mistakes have been noticed that should be removed before further processing. I recommend it for publication after thoroughly going through to eliminate all formatting mistakes.    

Author Response

Reviewer 1

The manuscript is well structured and organized. The solubility and in turn bioavailability of vardenafil was improved by making complexes of vardenafil with three sweetening agents. The lab scale findings could be used for future prospective to go in this direction. The results are systematically presented and well narrated. However, few formatting mistakes have been noticed that should be removed before further processing. I recommend it for publication after thoroughly going through to eliminate all formatting mistakes.    

The authors would like to thanks the reviewer for the encouraging and positive comments. The authors have been through the manuscript to correct manuscript formatting.

Reviewer 2 Report

  1. why it has been prepare in SD then converted to dispersible film.
  2. the rationale of study missing.
  3. Authots have used old carrier system to prepare SD. Sweeting agent have already reported drawbacks.
  4. They must select the new carrier system to get better result.
  5. Discussion is poor. It must be supported with cited literature.
  6. the pharmacokinetic profile seems to be similar at most time point. There is significant difference acheived at most time point.
  7. Conclusion must be short and give important findings of the study.
  8. Statistical data missing
  9.  

Author Response

Reviewer 2

  1. why it has been prepare in SD then converted to dispersible film.

Our hypothesis was that solid dispersion was prepared first to improve water solubility of the drug and then load them on the film matrix. This hypothesis was workable from solubility studies and DSC and FTIR.

  1. the rationale of study missing.

The design of formulation was to improve solubility and dissolution rate of vardenafil and loading it on different types of hydrophilic polymers commonly used in film formulations such as methyl cellulose, sodium, alginate, poly vinyl pyrrolidone to fabricate ODFs for spontaneous and rapid effects for treatment of erectile dysfunction.

  1. Authots have used old carrier system to prepare SD. Sweeting agent have already reported drawbacks.

This is a valid point sweetening agents used have been around but we believe for the first time we employed them in coprecipitated mixtures with an insoluble drug. Other carriers such as cyclodextrins and HPMC did not produce any better results than we obtained with the sweetening agents.

  1. They must select the new carrier system to get better result.

We appreciate that and future studies will look into investigating novel carriers.

  1. Discussion is poor. It must be supported with cited literature.

While we could wish the reviewers to highlight the paragraphs needs citation but we have been through the whole manuscript to add references whenever needed.

  1. the pharmacokinetic profile seems to be similar at most time point. There is significant difference acheived at most time point.

As mentioned in the literature, vardenafil absorption is highly affected by the presence of food. In this study, the volunteers were fasted to eliminate these differences and it has now been modified accordingly. Further a steep and rapid absorption peak with minimal fluctuations were obtained especially as it was observed with the ODF.

  1. Conclusion must be short and give important findings of the study.

The conclusion section has now been modified accordingly.

  1. Statistical data missing

Statistical data have been provided whenever possible.

Reviewer 3 Report

Dear authors,

your manuscript is really interesting and presents state of art in terms of dosage form development. It is easy to follow and understand. Nonetheless, I have a few concerns about methodology as well obtained results:

1. From  FDA recommendation perspective, your compositions are hard to be named as ODFs whereas disintegration time for almost all of them is more than 180 seconds. Moreover, it is recommended for orally disintegrating formulations to disintegrate in time below 30 seconds.

2. Why author decided to test dissolution in 500 mL wheres recommended by FDA method for vardenafil ODT is like below? To make any further comparison with tablets or ODTs we should have comparable conditions.

3. It is much more convenient from a dissolution and bioavailability perspective to compare ODF to ODT formulations as we have that introduced into the market. 

Best regards,

PS. There is a typo in line 90: should be ODF.

Author Response

Reviewer 3

your manuscript is really interesting and presents state of art in terms of dosage form development. It is easy to follow and understand. Nonetheless, I have a few concerns about methodology as well obtained results:

  1. From  FDA recommendation perspective, your compositions are hard to be named as ODFs whereas disintegration time for almost all of them is more than 180 seconds. Moreover, it is recommended for orally disintegrating formulations to disintegrate in time below 30 seconds.

Disintegration time for oral film formulation is highly dpendent on the methods used. In vitro disintegration time obtained with this method was only 120 seconds for F8. In vivo disintegration during clinical study was fater and patients did not show any inconvenience or foreign undissolved matters.

  1. Why author decided to test dissolution in 500 mL wheres recommended by FDA method for vardenafil ODT is like below? To make any further comparison with tablets or ODTs we should have comparable conditions.

Vardenafil HCl

Tablet (Orally Disintegrating) 

II (Paddle) 

50

0.1 N HCl 

900

5, 10, 15 and 30 

05/15/2014

We run preformulation investigations and based on drug solubility. A volume of 500 ml is sufficient to secure sink conditions.

  1. It is much more convenient from a dissolution and bioavailability perspective to compare ODF to ODT formulations as we have that introduced into the market. 

This is a valid point. The aim of this paper to compare ODF with the most widely prescribed tablet forms to compare bioavailability differences. Future investigation will include ODT onboard.

Best regards,

  1. There is a typo in line 90: should be ODF.

The typo has now been corrected.

Round 2

Reviewer 2 Report

Accept

Reviewer 3 Report

Dear authors,

thank you for your responses. The manuscript presents valuable data for publication.

This manuscript is a resubmission of an earlier submission. The following is a list of the peer review reports and author responses from that submission.

Round 1

Reviewer 1 Report

The Authors present development and in vivo study of oral dispersible films (ODF) with vardenafil. The aim of the study is clear and the construction of the paper is very good. My main concerns are as follows:

1/ Six out of the eight developed and studied formulations contain methylcellulose and disintegration time of these films is too long (5-11 min). It was a bad choice of this polymer and this could be verified already a priori, on the basis of the literature or products on the market. It is not true what the Authors write in the Introduction that ODFs should disintegrate in oral cavity within few minutes. Finally the experiments performed with these six formulations do not have sufficient value, neither practical nor scientific. It would be enough to present only one formulation with MC. The Authors should present a clear explanation why most of the work was done on the formulations which do not have potential to be used as ODF.

2/ The in vivo experiments gave the results which are at least surprising – the blood levels of vardenafil in 3 tested volunteers are practically the same with practically no deviations (RSD for all pharmacokinetic parameters was 1-2%), while pharmacokinetics of this drug presented in the literature is poorly reproducible and that was also the reason why the Authors tried to develop a new formulation, indicating that “erratic oral absorption …. is the main drawback of vardenafil therapy”. The Tmax was 120 min while in the literature 60 min is given, but at the same time the Authors conclude that ODF “produced rapid plasma peaking”.

3/ In vivo experiments were also performed to study the taste. However only the relative sweeteness and bitternes is reported, while the conclusion whether any formulation is acceptable has not been provided. No information is given whether this experiment got the approval of the Ethical Commission. Why the in vivo disintegration time was not studied in the same experiment? The volunteers kept the ODF in the mouth up to 10 min.

Some minor comments are the following:

  • 12 is not a good reference for demonstrating that ODF allows to reduce first-pass metabolism; actually this effect does not take place as a rule because the drug absorption occurs from the GI tract not from the mouth.
  • No information is given if the drug was dissolved in the final formulation or suspended
  • Table 1 – is this a composition of wet or dry film (10 g is a quantity of polymer solution?). Glycerol, the main constituent is not characterized in the Materials section
  • Blood sample preparation is unclear (standard solution, 50 ml flask with no solvent, etc.)
  • Lines 230-231 – sentence incomplete
  • FTIR spectra should be superimposed on the graphs what allows for comparison
  • Figures 7 and 8 are too large

Author Response

Reviewer 1

The Authors present development and in vivo study of oral dispersible films (ODF) with vardenafil. The aim of the study is clear and the construction of the paper is very good. My main concerns are as follows:

1/ Six out of the eight developed and studied formulations contain methylcellulose and disintegration time of these films is too long (5-11 min). It was a bad choice of this polymer and this could be verified already a priori, on the basis of the literature or products on the market. It is not true what the Authors write in the Introduction that ODFs should disintegrate in oral cavity within few minutes. Finally the experiments performed with these six formulations do not have sufficient value, neither practical nor scientific. It would be enough to present only one formulation with MC. The Authors should present a clear explanation why most of the work was done on the formulations which do not have potential to be used as ODF.

The design of formulation was to study different types of hydrophilic polymers commonly used in film formulations such as methyl cellulose, sodium, alginate, poly vinyl pyrrolidone on the disintegration time and come up with the most suitable formulations.  We believe there are scarce reports on studying the effect of these polymers in different ration with the film forming polymer PVP on the disintegration time. It would be useful for the reader to be informed on such pros and cons of these commonly used polymers.The organoleptic properties and the effect of sweetening agents has been also investigated.

The introduction section has now been modified accordingly. Lines 91-95.

2/ The in vivo experiments gave the results which are at least surprising – the blood levels of vardenafil in 3 tested volunteers are practically the same with practically no deviations (RSD for all pharmacokinetic parameters was 1-2%), while pharmacokinetics of this drug presented in the literature is poorly reproducible and that was also the reason why the Authors tried to develop a new formulation, indicating that “erratic oral absorption …. is the main drawback of vardenafil therapy”. The Tmax was 120 min while in the literature 60 min is given, but at the same time the Authors conclude that ODF “produced rapid plasma peaking”.

It is a valid point. As mentioned in the literature, vardenafil absorption is highly affected by the presence of food. In this study, the volunteers were fasted to eliminate these differences and it has now been modified accordingly. Further a steep and rapid absorption peak was obtained especially as it was observed with the ODF.

3/ In vivo experiments were also performed to study the taste. However only the relative sweeteness and bitternes is reported, while the conclusion whether any formulation is acceptable has not been provided. No information is given whether this experiment got the approval of the Ethical Commission. Why the in vivo disintegration time was not studied in the same experiment? The volunteers kept the ODF in the mouth up to 10 min.

This study was approved by Ehtics Approval Committee and the number was provided. Modified Huan Gustatory Taste is one of the well-established methods to evaluate the organoleptic properties. The relative sweetness was measured against validated standard references controls 

Some minor comments are the following:

  • 12 is not a good reference for demonstrating that ODF allows to reduce first-pass metabolism; actually this effect does not take place as a rule because the drug absorption occurs from the GI tract not from the mouth.
  • A more suitable reference has now been provided.
  • No information is given if the drug was dissolved in the final formulation or suspended

The section has now been modified accordingly to indicate that the drug was dissolved in the final polymer solution.

  • Table 1 – is this a composition of wet or dry film (10 g is a quantity of polymer solution?). Glycerol, the main constituent is not characterized in the Materials section

The concentration of the drug was only based on the drug weight. Glycerol supplier has now been provided.

  • Blood sample preparation is unclear (standard solution, 50 ml flask with no solvent, etc.)

The section has now been modified accordingly.

  • Lines 230-231 – sentence incomplete

The sentence has now been corrected.

  • FTIR spectra should be superimposed on the graphs what allows for comparison

FTIR spectra were added separately to improve the resolution.

  • Figures 7 and 8 are too large

The size of Figures 7 and 8 has now been reduced.

Reviewer 2 Report

The manuscript on vardenafil oral dispersible films is interesting and well written.  The following issues should be addressed:

1) During preparation of the solid dispersed mixtures, the solutions were heated to 60 degree C for complete evaporation.  How did this elevated temperature affect the stability of vardenafil?

2) Please give more information on the solubility studies, namely how much powder was added to how much distilled water in order to ensure a supersaturated solution would be maintained?

3) Was the selectivity of the UV spectrophotometric method used in the solubility study established in order to ensure that the other components (sugars) did not interfere with the UV absorption of the vardenafil?

4) The quantity of vardenafil in each ODF formulation is listed at 120 mg, however, it is further stated that its content was kept at 10% w/w, please clarify.  Based on this, how was it ensured each volunteer in the pharmacokinetic study received exactly 10 mg vardenafil in each ODF? Was the ODF cut to the correct size based on the content assay?

5)  A major concern is the gustatory sensation test.  Was ethics approval also obtained for this study since human subjects were used to score the taste - please provide the ethics number for this part of the study in addition to what was given for the pharmacokinetic study.  Each volunteer in this test received an ODF of all 8 formulations with only 20 min washout periods in between.  How did the authors ensured that these volunteers did not get an overdose of vardenafil since each formulation was kept in the mouth for 10 min before gargling was done with water?  Were these volunteers also monitored with respect to blood pressure etc.

Author Response

Reviewer 2

The manuscript on vardenafil oral dispersible films is interesting and well written.  The following issues should be addressed:

  • During preparation of the solid dispersed mixtures, the solutions were heated to 60 degree C for complete evaporation.  How did this elevated temperature affect the stability of vardenafil?

Vardenafil is chemically stable enough to withstand the preparation condition (data yet to publish).

  • Please give more information on the solubility studies, namely how much powder was added to how much distilled water in order to ensure a supersaturated solution would be maintained?

The section has now been modified to add the required information.

  • Was the selectivity of the UV spectrophotometric method used in the solubility study established in order to ensure that the other components (sugars) did not interfere with the UV absorption of the vardenafil?

The method was validated and no interference was detected from the used sugars.

  • The quantity of vardenafil in each ODF formulation is listed at 120 mg, however, it is further stated that its content was kept at 10% w/w, please clarify.  Based on this, how was it ensured each volunteer in the pharmacokinetic study received exactly 10 mg vardenafil in each ODF? Was the ODF cut to the correct size based on the content assay?

There is no contradiction, the total amount based on dry weight of polymer dissolved and other excipients (plasticizers). Based on this, a film with a diameter of 1 to 1.2 cm diameter cut out by cork borer resulting in a film with average drug content of 10 mg.

  • A major concern is the gustatory sensation test.  Was ethics approval also obtained for this study since human subjects were used to score the taste - please provide the ethics number for this part of the study in addition to what was given for the pharmacokinetic study.  Each volunteer in this test received an ODF of all 8 formulations with only 20 min washout periods in between.  How did the authors ensured that these volunteers did not get an overdose of vardenafil since each formulation was kept in the mouth for 10 min before gargling was done with water?  Were these volunteers also monitored with respect to blood pressure etc.

The patient was instructed to swallow but spit the ODF after giving the score. Ethics approval number has been provided.

Round 2

Reviewer 1 Report

Pharmacokinetic data should be further discussed. The use of MC is not enough justified.

Author Response

Reviewer 1

Pharmacokinetic data should be further discussed.

The pharmacokinetics section in the results and discussion has now been modified accordingly

The use of MC is not enough justified.

The justification of MC and other polymers (SA and PVP) has now been highlighted in the introduction section.